# H-NeRF: Neural Radiance Fields for Rendering and Temporal Reconstruction of Humans in Motion

**Hongyi Xu**
Google Research
hongyixu@google.com

**Thiemo Alldieck**
Google Research
alldieck@google.com

**Cristian Sminchisescu**
Google Research
sminchisescu@google.com

## Abstract

We present neural radiance fields for rendering and temporal (4D) reconstruction of humans in motion (H-NeRF), as captured by a sparse set of cameras or even from a monocular video. Our approach combines ideas from neural scene representation, novel-view synthesis, and implicit statistical geometric human representations, coupled using novel loss functions. Instead of learning a radiance field with a uniform occupancy prior, we constrain it by a structured implicit human body model, represented using signed distance functions. This allows us to robustly fuse information from sparse views and generalize well beyond the poses or views observed in training. Moreover, we apply geometric constraints to co-learn the structure of the observed subject – including both body and clothing – and to regularize the radiance field to geometrically plausible solutions. Extensive experiments on multiple datasets demonstrate the robustness and the accuracy of our approach, its generalization capabilities significantly outside a small training set of poses and views, and statistical extrapolation beyond the observed shape.

## 1 Introduction

Enabling free-viewpoint video of a human in motion, based on a sparse set of views, is extremely challenging, but has many applications. Our work is motivated by a breadth of transformative 3D use cases, including immersive visualization of photographs, virtual clothing and try-on, fitness, as well as AR and VR for improved communication or collaboration. However, so far static scenes and rigid objects have been the primary subject of research. In pursuing realistic novel-view synthesis two schools of thought have been established: 1) 3D reconstruction methods aim to recover the geometry of the observed scene as accurately as possible, with novel views generated using classical rendering pipelines [30, 42]. 2) Image-based rendering techniques [7, 10, 40] and very recently neural radiance fields [29] primarily aim at image production quality without explicitly constructing an accurate 3D geometric model. While these techniques explicitly or implicitly reconstruct the scene geometry sometimes, this is however not guaranteed to accurately resemble the true scene geometry. We argue that novel-view rendering and reconstruction are two sides of the same coin, and that reliable viewpoint generalization, especially given relatively few input views, would require good quality for both. To this end, we propose a unified model in order to support both robust reconstruction and photo-realistic rendering. Dynamic scenes, especially those capturing a human in motion, add considerable complexity to the problem: while static scenes can be observed from many views by a camera moving through the scene, any configuration of a dynamic scene is typically observed only from sparse views. Moreover, the scene geometry and its appearance may change considerably over time. To cope with few views, some methods integrate scene knowledge over time by warping observations into a common reference frame [35, 37]. At test time, the information is warped back to the desired state and rendered from a novel view. Extrapolating to unseen motion, however, remains challenging. For scenes capturing people, this means that only poses seen during training can be rendered at test time. For some applications, however, rendering the subject over a broad range of motions, or in novel poses, is desirable. To make generalisation over poses and views possible, we rely on additional problem domain knowledge in the form of a human body model, imGHUM [4]. imGHUM is an implicit signed distance function (SDF) conditioned on generative shape and pose

35th Conference on Neural Information Processing Systems (NeurIPS 2021).

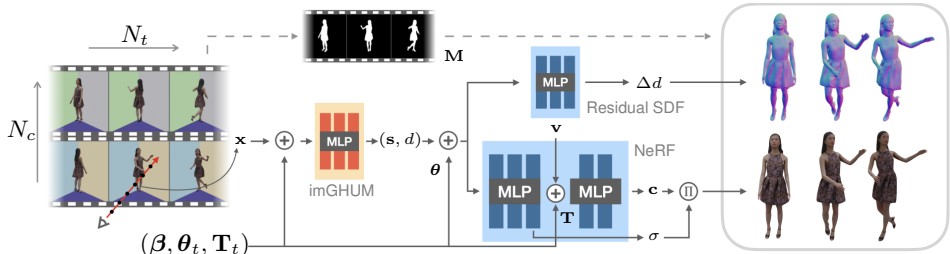

Figure 1: Overview of H-NeRF. Given a set of images of a human performer collected from a sparse set of calibrated cameras, we train a geometry-aware neural radiance field by co-learning a deformable signed distance field. First, we estimate the body shape $\beta$ and track the articulated pose $\theta, \mathbf{T}$ using the implicit statistical human body model imGHUM (orange). imGHUM encodes all 3D spatial points $\mathbf{x}$ across frames with a 4D point descriptor $(\mathbf{s}, d)$ referencing a canonical frame. Using the foreground mask $\mathbf{M}$, we co-train a residual SDF and a NeRF network in that canonical space, in order to integrate all image observations into a consistent implicit 3D geometry $\Delta d$ and its view-depended ($\mathbf{v}$) radiance representation $(\mathbf{c}, \sigma)$. The trainable residual SDF and NeRF networks (in blue) are conditioned on the body pose $\theta$ and the root transformation $\mathbf{T}$ to model pose-dependent geometric and appearance variations. Our framework supports both accurate 3D geometric reconstruction and free-viewpoint rendering, and generalizes well to novel views, shapes, and poses.

codes learned from a large corpus of dynamic human scans. In this work, imGHUM is used as the common reference frame for a neural radiance field and as a structured prior for robust reconstruction. Additionally, since imGHUM can represent a broad distribution of statistically valid human poses and shapes, we can render the reconstructed subject in novel poses and even with modified body shapes. In summary, our system supports photo-realistic free-view point temporal rendering of a human subject given only sparse camera observations. By conditioning on an implicit human body model, we can render considerably different viewpoints, body poses, and body shapes compared to those observed in training. Our carefully designed losses ensure not only good image quality but also plausible temporal reconstructions that can be used for the free-viewpoint visualisation of human performance capture.

## 2 Related Work

**Human performance capture** is the process of reconstructing the dynamic (4D) geometry of a human subject as observed in one or multiple synchronized views. Pioneering methods deform a rigged template mesh against silhouettes observed in multiple views [6, 50], estimated skeleton key-points [12], or image correspondences [9]. Later systems relying on RGB-D video streams [5] or monocular capture [3, 2, 17, 52, 16, 15] have also been presented. All these methods rely either on a pre-scanned template mesh or on a body model that is deformed to explain the image evidence.

**Neural representation for view synthesis.** With the advent of neural networks, researchers have begun to explore alternative solutions to represent a scene in order to support novel view synthesis and photo-realistic rendering; we refer the reader to [48] for a survey. Different scene representation have been explored. Some methods use voxel grids to embed the scene [23, 45]. For rendering, the voxel grid is probed by shooting a ray and by linearly interpolating voxel values. These samples are then transformed into color values using a neural network. Others have proposed neural textures [49, 43] that can be rendered based on view-dependent effects or texture synthesis networks [22] to realistically render meshes. In a similar spirit, other methods render point clouds, where each point carries local appearance information [26, 1]. Riegler and Koltun [40] reproject features from nearby views and rely on a SfM reconstruction scaffold for novel view synthesis. With the recent advent of implicit function networks [25, 34, 8, 27], such 3D representations have been explored for rendering and view synthesis with great success. In contrast to discrete voxel representations, textures, or point clouds, these methods represent the scene as a continuous function and thus are not bound to a specific image or volume resolution. In the pioneering work of Sitzmann et al. an implicit function produces features displayed using a neural renderer [46]. Follow up methods focus on 3D geometric reconstruction [44], and use 2D supervision [31, 53].

**Neural Radiance Fields** (NeRFs) are a recent approach to represent scenes for novel view synthesis. Mildenhall et al. [29] introduce Neural Radiance Fields resented as fully connected neural networks,

where the input is a spatial query point and a viewing direction. The output is a volume density and the emitted radiance at the query location in the direction of the viewer. By ray-tracing using this simple representation, one can generate photo-realistic images from novel views. Despite the excellent quality of results, one drawback is the slow rendering time. To this end, researchers have presented faster versions that e.g. transform the radiance field into more efficient sparse grids [18] or remove the dependence on viewing-direction during rendering by estimating a spherical harmonic representation of the radiance function [54]. Others improve fidelity or rendering time by tackling ambiguities in the original formulation [56], spatial decomposition into multiple NeRFs [39], or combining NeRF with sparse voxel fields [21]. Initial work on adapting NeRF to dynamic scenes has been presented as well. Park et al. [35] produce "Nerfies" (NeRF-Selfies) from videos where subjects carefully move a camera around their head. The scene information is fused by warping query points into a canonical reference frame. Similarly, Pumarola et al. [37] produce dynamic NeRFs from synthetic animation data. Related to our approach, some methods integrate human body models to fuse information over time. A-NeRF [47] uses a skeleton to rigidly transform NeRF features to refine estimated 3D poses. A similar approach is followed in NARF [32] for view synthesis. Most related, Neural Body [36] attaches learnable features to the vertices of a SMPL body model [24]. These features are processed using a sparse 3D convolutional network, where the output forms a neural radiance field. In contrast to our approach, the resolution is bounded by the spatial resolution of the 3D convolutional network and no geometric supervision is used. We highlight differences in §5.

## 3 Background

Given a collection of images capturing a dynamic scene of a human in motion, observed from a sparse set of calibrated camera views (in the limit a monocular camera, as we will show), we aim to learn both the detailed temporal geometry of the human in motion, and to render the sequence from novel camera views and for different human poses. To this end, our work unifies two main methodologies: 1) implicit 3D human representations, and 2) volumetric radiance fields. In this section, we provide the relevant background on both representations, which we co-learn in a joint framework.

**Neural Radiance Fields.** A neural radiance field (NeRF) [29] represents a 3D scene as a continuous function of color volume densities. The model consists of a neural network function $F_\omega$ that maps a 3D spatial point $\mathbf{x} \in \mathbf{R}^3$ and a viewing direction $\mathbf{v} \in \mathbf{R}^3$ to a volume density $\sigma \in \mathbf{R}^+$ and a radiance $\mathbf{c}(\mathbf{x}, \mathbf{v}) \in \mathbf{R}^3$ emitted towards the viewer. In practice, NeRF encodes the inputs $\mathbf{x}$ and $\mathbf{v}$ using a sinusoidal positional encoding $\boldsymbol{\gamma} : \mathbf{R}^3 \rightarrow \mathbf{R}^{3+6m}$ that projects a coordinate vector into a high-dimensional space using a set of sine and cosine functions of $m$ increasing frequencies. Given a ray $\mathbf{r} = \mathbf{o} + s\mathbf{v}$ with $N$ samples $\{\mathbf{x}\}$ originating from a camera location $\mathbf{o}$, NeRF integrates radiance values along the ray by means of alpha blending. The pixel/ray color is approximated with numerical quadrature [33],

$$\mathbf{C}(\mathbf{r}) = \sum_{i=1}^{N} \alpha(\mathbf{x}_i) \prod_{j<i}(1 - \alpha(\mathbf{x}_j))\mathbf{c}(\mathbf{x}_i, \mathbf{v}), \tag{1}$$

$$\alpha(\mathbf{x}_i) = 1 - \exp\big(-\sigma(\mathbf{x}_i)\delta_i\big), \tag{2}$$

where $\alpha(\mathbf{x}_i)$ is the transparency by accumulating transmittance along the ray, and $\delta_i = |\mathbf{x}_{i+1} - \mathbf{x}_i|$ is the distance between adjacent samples. The NeRF function $F_w$ is fully differentiable and its network parameters $w$ can be optimized using an image reconstruction loss [29]. An approximate 3D scene geometry $\mathbf{S}_F = \{\mathbf{x}|\sigma(\mathbf{x}) = \sigma_h\}$ can be extracted from the trained opacity field via Marching Cubes [20] at a density threshold $\sigma_h$.

**Implicit Generative Human Models.** Implicit human body surfaces are typically represented as the decision boundary of either binary occupancy classifiers [11, 41, 28] or signed distance functions [14, 4]. Specifically, our work builds upon the SOTA statistical implicit human model imGHUM [4] $H_\omega : (\mathbf{T}^{-1}\mathbf{x}, \boldsymbol{\beta}, \boldsymbol{\theta}) \rightarrow (d, \mathbf{s})$ that maps a 3D spatial point $\mathbf{x}$, unposed with root joint transformation $\mathbf{T} \in \mathbf{R}^{4\times3}$, to its signed distance value $d \in \mathbf{R}$ with respect to a body surface parameterized with body shape $\boldsymbol{\beta} \in \mathbf{R}^{16}$ and articulated pose $\boldsymbol{\theta} \in \mathbf{R}^{118}$. In addition to $d$, imGHUM returns implicit continuous semantics $\mathbf{s} \in \mathbf{R}^3$ of the query point which corresponds to the 3D coordinate of the nearest surface point defined on a canonical surface. We refer to the original paper [4] for details. Essentially, imGHUM builds a human-centric 4D semantic descriptor $(d, \mathbf{s})$ for all spatial points in the neighborhood of the surface. imGHUM is trained with a large collection of human scans of diverse body shapes and poses, sharing the same generative shape and pose latent

code with the mesh-based statistical human model GHUM [51]. The implicit articulated 3D human body $\mathbf{S}_H(\boldsymbol{\beta}, \boldsymbol{\theta}, \mathbf{T}) = \{\mathbf{x}|d(\mathbf{x}) = 0\}$ is defined by the the zero-isosurface of the signed distance field.

# 4 Method

We present the details of our main contribution, H-NeRF, a novel neural network (fig. 1) that exploits the power of volumetric radiance fields to learn complex human structure and appearance, by relying on statistical implicit human pose and shape signed distance functions for accurate geometric reconstruction. Further, H-NeRF relies on imGHUM, an implicit model of articulated human pose and shape, as a rich geometric prior, to integrate scene information over time, and to represent human articulation. Co-learning both a radiance field and a signed distance function of scene geometry consistently in a unified framework, enables accurate 3D geometric reconstruction and volume rendering of a dynamic human in motion, through 1) consistent integration of image observations over time, and 2) good generalization capability for novel viewpoints, human poses, and for statistically extrapolated shapes, given only very few training poses and camera views.

Given a video of a human in motion, observed by a sparse set of calibrated cameras, our objective is to generate free-viewpoint video of the observed person and to reconstruct the underlying 4D geometry. The set of input images, with a resolution of $w \times h$ pixels, is denoted as $\{\mathbf{I}_t^c \in \mathbf{R}^{w \times h \times 3}|c = 1, \ldots, N_c, t = 1, \ldots, N_t\}$, where $c$ is the camera index, $N_c$ is the number of cameras, $t$ is the frame index, and $N_t$ is the number of frames. For each image, we apply [13] to obtain the binary foreground human mask $\mathbf{M}_t^c \in \mathbf{R}^{w \times h}$. In addition, we obtain a temporally consistent imGHUM latent shape and pose codes, $\boldsymbol{\beta}$ and $(\boldsymbol{\theta}_t, \mathbf{T}_t)$, respectively, at each frame index $t$, by optimizing an imGHUM-equivalent parametric replica under multi-view keypoint and body segmentation losses [55]. We refer to our Sup. Mat. for details on the imGHUM fitting process.

In the sequel we first explain our adaptations to NeRF for the static case. imGHUM is used as a prior in order to bias the reconstruction of the observed scene towards a more accurate human geometry. We continue by explaining the additional methodological innovation needed in the dynamic case. Hereby, imGHUM provides spatial-temporal correspondences and is used as the common reference frame to fuse information across different views and time instances.

## 4.1 Static Semantic Human NeRF

We first formulate H-NeRF for a human capture at a single moment in time ($N_t = 1$). From multi-view image observations, we co-learn a radiance field $F_\omega : \mathbf{x}, \mathbf{v} \rightarrow (\mathbf{c}, \sigma)$ for free-viewpoint rendering, and a signed distance function $\hat{H}_\omega : \mathbf{x} \rightarrow \hat{d}$ for 3D geometric reconstruction. We use $\hat{H}_\omega$ for the dressed subject in order to distinguish it from the body's imGHUM SDF $H_\omega$.

Like NeRF, we formulate an $L_1$ image reconstruction loss to optimize $F_w$ as

$$\mathcal{L}_{\text{rec}} = \sum_{\mathbf{r} \in \mathcal{R}} \|\bar{\mathbf{C}}(\mathbf{r}) - \mathbf{C}(\mathbf{r})\|_1, \tag{3}$$

where $\mathcal{R}$ denotes the batched set of all pixels/rays, and $\bar{\mathbf{C}}(\mathbf{r}), \mathbf{C}(\mathbf{r})$ are observed and rendered pixel colors, cf. (1), respectively. However, under sparse training camera views, the volumetric radiance field is not well regularized, leading to poor generalization to novel viewpoints, cf. fig. 2. Specifically, we observe that the model encounters difficulties in correctly representing the scene and in separating the person from the background. The model fails to learn a semantically meaningful opacity field, i.e. $\alpha = 0$ in free space, and 1 if occupied.

**Coarse Scene Structuring.** Using imGHUM fits for all training images, we can spatially locate the person in 3D. We define a 3D bounding box $\mathbf{B} \in [\underline{\mathbf{S}_H} - \epsilon, \overline{\mathbf{S}_H} + \epsilon]$ around the detected person, where $\underline{\mathbf{S}_H}, \overline{\mathbf{S}_H}$ are the minimal and maximal coordinates of the human body surface $\mathbf{S}$ and $\epsilon$ is a spatial margin reserved for geometry not modeled by imGHUM. All radiance points associated to the rendering of the person should reside inside the bounding box, leading to a 3D segmentation loss

$$\tilde{\mathbf{C}}(\mathbf{r}) = \sum_{i=1}^{N} b(\mathbf{x}_i)\alpha(\mathbf{x}_i) \prod_{j<i}(1 - b(\mathbf{x}_j)\alpha(\mathbf{x}_j))\mathbf{c}(\mathbf{x}_i, \mathbf{v}), \tag{4}$$

$$\mathcal{L}_{\text{mask}} = \sum_{\mathbf{r} \in \mathcal{R}} \left\|\mathbf{M}(\mathbf{r})(\bar{\mathbf{C}}(\mathbf{r}) - \tilde{\mathbf{C}}(\mathbf{r}))\right\|_1, \tag{5}$$

where $b(\mathbf{x}_i)$ is 0 if outside of $\mathbf{B}$ and 1 otherwise, and $\mathbf{M}(\mathbf{r})$ the image mask. We rely on the mask, so the loss is only applied to image observations from the subject, and not the remaining scene geometry.

**Unifying SDF with NeRF.** After structuring the scene coarsely, we now couple the estimated radiance field with an implicit signed distance-based 3D reconstruction, in order to further regularize the opacity distribution. A signed distance function associated to the detected person in the image naturally comes with a 3D classifier for all spatial points, where $\hat{d}(\mathbf{x}_i) > 0$ means $\mathbf{x}_i$ is in free space, whereas $\mathbf{x}_i$ lies within the subject when $\hat{d}(\mathbf{x}_i) <= 0$. We rely on this insight in order to co-learn a SDF of the performer and constrain the radiance field. To this end, we introduce a pseudo alpha value $\dot{\alpha}(\mathbf{x}_i) = \phi(\gamma\hat{d}(\mathbf{x}_i))$ where $\phi$ is a Sigmoid activation function and $\gamma$ controls the sharpness of the boundary. To refine the NeRF opacity semantics, especially for the volume in the neighborhood of the human surface $\mathbf{B}$, we formulate two losses

$$\hat{\mathbf{C}}(\mathbf{r}) = \sum_{i=1}^{N} \hat{\alpha}(\mathbf{x}_i)\prod_{j<i}(1-\hat{\alpha}(\mathbf{x}_j))\mathbf{c}(\mathbf{x}_i,\mathbf{v}), \quad \hat{\alpha}(\mathbf{x}_i) = b(\mathbf{x}_i)\dot{\alpha}(\mathbf{x}_i) + (1-b(\mathbf{x}_i))\alpha(\mathbf{x}_i), \quad (6)$$

$$\mathcal{L}_{\text{blend}} = \sum_{\mathbf{r}\in\mathcal{R}}\left(\|\mathbf{M}(\mathbf{r})\big(\bar{\mathbf{C}}(\mathbf{r}) - \hat{\mathbf{C}}(\mathbf{r})\big)\|_1 + \eta\|(1-\mathbf{M}(\mathbf{r}))\big(\bar{\mathbf{C}}(\mathbf{r}) - \hat{\mathbf{C}}(\mathbf{r})\big)\|_1\right), \quad (7)$$

$$\mathcal{L}_{\text{geom}}(\mathbf{r}) = \sum_{i=1}^{N} b(\mathbf{x}_i)\text{BCE}\big(\phi\big(\lambda(\sigma_h - \sigma(\mathbf{x}_i))\big), \dot{\alpha}(\mathbf{x}_i)\big), \quad (8)$$

where $\hat{\mathbf{C}}(\mathbf{r})$ is the rendered pixel color with blended alpha values $\hat{\alpha}$ replacing NeRF alpha values $\alpha$ with SDF-based pseudo alpha $\dot{\alpha}$ for all ray points inside the bounding box $\mathbf{B}$. The first term in $\mathcal{L}_{\text{blend}}$ requires the rendered pixel color for all the intersecting rays within the human mask to come from the surface, whereas the second term assumes that all background color is formed from ray samples outside of $\mathbf{B}$. We set $\eta = 1$ when no other geometry than the person is inside $\mathbf{B}$ and tune $\eta$ down if the assumption is violated (e.g. person standing on a floor). The term $\mathcal{L}_{\text{geom}}$ uses the binary cross entropy loss to couple the NeRF surface boundary with the zero-isosurface of the signed distance function describing the subject. While the coupling terms given by (7) and (8) act as strong priors for the opacity distribution, during test time, we still rely on volumetric radiance rendering with learned NeRF alpha values $\alpha$ to support transparency effects and complex geometry for structures like hair.

**Image-based SDF learning.** Learning the signed distance field $\hat{H}_\omega$ from scratch using a sparse set of training images is still challenging. The model often fails to reconstruct reasonable human geometry, even when using coupling losses (7) and (8). We therefore leverage imGHUM as an inner layer for the target human reconstruction and combine it with a light-weight residual SDF network $\Delta H_\omega : \mathbf{x} \to \Delta d$. The residual SDF models surface details, including hair and clothing, that are not represented by imGHUM. The final signed distance for $\mathbf{x}_i$ becomes $\hat{d}(\mathbf{x}_i) = d(\mathbf{x}_i|\boldsymbol{\beta}, \boldsymbol{\theta}, \mathbf{T}) + \Delta d(\mathbf{x}_i)$. Given the training images, we learn the personalized residual SDF using our coupling losses, given by (7) and (8), and additionally apply geometric regularization

$$\mathcal{L}_{\text{seg}} = \sum_{\mathbf{r}\in\mathcal{R}}\text{BCE}(\mathbf{M}(\mathbf{r}), \hat{d}_{\text{min}}(\mathbf{r})), \quad (9)$$

$$\mathcal{L}_{\text{eik}}(\mathbf{r}) = \sum_{i=1}^{N} b(\mathbf{x}_i)(\|\nabla_{\mathbf{x}_i}\hat{d}(\mathbf{x}_i)\|_2 - 1)^2, \quad \mathcal{L}_{\text{reg}} = \|\psi(\Delta d)\|_1, \quad (10)$$

where $\hat{d}_{\text{min}}(\mathbf{r})$ denotes the minimal signed distance for all sampled ray points and $\psi$ is the ReLU activation function. The term $\mathcal{L}_{\text{seg}}$ ensures that if a pixel is inside the human segmentation mask, there should be at least one intersection between the ray and the 3D human surface and therefore $\hat{d}_{\text{min}}(\mathbf{r})$ should be non-positive. Otherwise, all ray samples should have positive signed distances. Using $\mathcal{L}_{\text{eik}}$, we enforce the composite SDF $\hat{d}$ to be approximately a signed distance function, i.e. incorporating Eikonal regularization [14]. The term $\mathcal{L}_{\text{reg}}$ regularizes the residual distance $\Delta d$ to be non-positive. This is because imGHUM should reside within the geometry and personalized geometric details given by $\Delta d$ should be modeled on top of the skin surface.

## 4.2 Dynamic Semantic Human NeRF

We now extend the framework to model dynamic human motion. To implicitly model scenes where the human moves, we learn a consistent, continuous function $G_\omega : (\mathbf{c}, \alpha, \hat{d}|\mathbf{z}) \rightarrow (\mathbf{c}', \alpha', \hat{d}'|\mathbf{z}')$ for both the geometry and the appearance variations across frames. For generalization to novel poses or shapes, $G_\omega$ should be conditioned on a semantically meaningful latent code $\mathbf{z}$ that can be interpolated and should ideally have good extrapolation properties. To integrate scene information over time in a single radiance field, we follow the approaches in [35, 37] and warp observations into a canonical reference frame. Given our constraints to $G_\omega$, imGHUM, conditioned on its semantic shape and pose latent code $(\boldsymbol{\beta}, \boldsymbol{\theta}, \mathbf{T})$, naturally provides spatial correspondences across frames. Given two spatial points $\mathbf{x}$ and $\mathbf{x}'$ in the space of two human instances $(\boldsymbol{\beta}, \boldsymbol{\theta}, \mathbf{T})$ and $(\boldsymbol{\beta}', \boldsymbol{\theta}', \mathbf{T}')$ respectively, we decide these are in correspondence if they have the same semantics and signed distances $(\mathbf{s}, d|\boldsymbol{\beta}, \boldsymbol{\theta}, \mathbf{T}) = (\mathbf{s}', d'|\boldsymbol{\beta}', \boldsymbol{\theta}', \mathbf{T}')$. Essentially, imGHUM assigns a 4D point descriptor $(\mathbf{s}, d) \in \mathbf{R}^4$ to any spatial point and deforms the volume continuously with respect to the parameterized articulated human body surface. Specifically, we apply $H_\omega : (\mathbf{T}^{-1}\mathbf{x}, \boldsymbol{\beta}, \boldsymbol{\theta}) \rightarrow (d, \mathbf{s})$ to map a spatial point $\mathbf{x}$ to a canonical point descriptor $(\mathbf{s}, d)$. We then modify the input to the NeRF function with the point descriptor as $F_w : (\mathbf{s}, d) \rightarrow (\mathbf{c}, \sigma)$, and similarly for the residual SDF function as $\Delta H_\omega : (\mathbf{s}, d) \rightarrow \Delta d$. imGHUM is pre-trained using a large corpus of 3D human scans, thus we keep it fixed in our network and directly use this prior to integrate structured geometric and appearance information across training frames. Given that the background scene is invariant to the human motion, we only apply the imGHUM warping function to spatial points inside the bounding box $\mathbf{B}$, thus largely improving computation time and lowering memory usage.

**Pose-Dependent Geometry and Appearance.** Training a single canonical NeRF and a residual SDF from a sequence integrates image observations into a consistent implicit geometric and volumetric representation of the radiance. However, we observe that fine-grained geometric and appearance variations caused by human motion are missing in the canonical NeRF and the residual SDF. To model such variations in geometry (clothing deformation) and appearance (lighting, self-shadows), we condition the NeRF function $F_\omega$ – the volume density $\sigma$ (geometry) and color value $\mathbf{c}$ (appearance) – as well as the residual SDF $\Delta H_\omega$ (geometry), on the body pose code $\boldsymbol{\theta}$. In addition, we also notice that the relative position and rotation of the person with respect to the scene largely affects appearance (due to illumination effects), but should not affect the geometry. Based on this observation, we additionally condition the NeRF color on the person's root transformation as $\mathbf{c}(\mathbf{x}|\boldsymbol{\theta}, \mathbf{T})$.

**H-NeRF Articulation.** Differently from prior work [35, 37], our H-NeRF modules are conditioned on semantic human latent codes $(\boldsymbol{\beta}, \boldsymbol{\theta}, \mathbf{T})$, leading to an SDF and radiance field that can be interpolated. Moreover, due to imGHUM's capability to accurately model geometric volume deformation for diverse human poses and shapes, H-NeRF enjoys strong generalisation to novel poses and even body shapes. Concretely, during inference, one can simply assign a different set of $(\boldsymbol{\beta}, \boldsymbol{\theta})$ to transfer the learned geometry and appearance to an unseen pose and shape configuration. For better generalization, we add Gaussian noise to the input code $(\boldsymbol{\theta}, \mathbf{T})$, preventing network overfitting to pose-dependent deformation and appearance effects experienced during training. When only a sparse set of camera views is available, We apply the same technique to NeRF, conditioned on viewing direction $\mathbf{v}$.

**Fine-tuning imGHUM Pose and Shape.** We observe that imGHUM fitting can be a source of error when learning $F_\omega$ and $\Delta H_\omega$ from dynamic image sequences. Instead of keeping parameters fixed, we further improve the fit during training by fine tuning a time-consistent shape correction $\Delta\boldsymbol{\beta}$ and per-frame pose correction $\Delta\boldsymbol{\theta}(t)$ with

$$\mathcal{L}_{\text{fit}} = \sum_{\mathbf{r} \in \mathcal{R}} \text{BCE}(\mathbf{M}(\mathbf{r}), d_{\min}(\mathbf{r}|\boldsymbol{\beta} + \Delta\boldsymbol{\beta}, \boldsymbol{\theta} + \Delta\boldsymbol{\theta}(t))), \quad \mathcal{L}_{\text{inc}} = \|\Delta\boldsymbol{\beta}\|_2 + \frac{1}{N_t}\sum_{t=1}^{N_t} \|\Delta\boldsymbol{\theta}\|_2, \quad (11)$$

where $\mathcal{L}_{\text{fit}}$ aligns the imGHUM fit with the human foreground segmentation and $\mathcal{L}_{\text{inc}}$ regularizes the incremental corrections.

## 5 Experiments

We quantitatively and qualitatively evaluate H-NeRF through ablations and by comparing with other methods. Please refer to our Sup. Mat. for additional results and ablation experiments. We first detail our model architecture, as well as the datasets and evaluation metrics used.

**Architecture and Training.** We use the same network architecture for all of our experiments, where the trainable modules $F_\omega$ and $\Delta H_\omega$ consist of eight 256-dimensional and six 128-dimensional MLPs

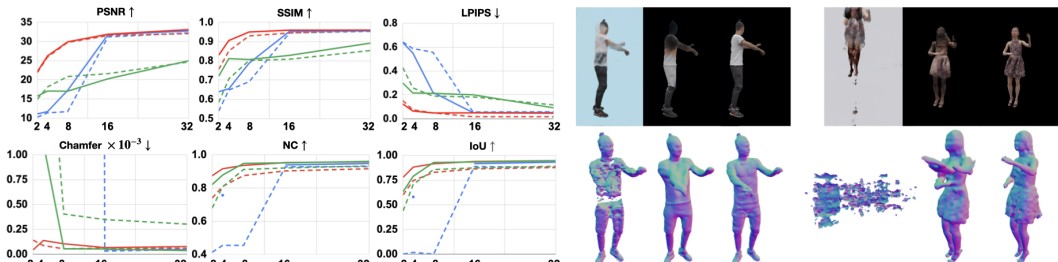

Figure 2: Left: H-NeRF (red, x-axis: #cameras) outperforms the original NeRF (blue) and IDR (green) in both novel view synthesis and 3D reconstruction of a static scene (16 test cameras). NeRF and IDR fail under sparse camera views (solid line: RenderPeople scan; dashed line: GHS3D scan). Right: we qualitatively show novel views and reconstructed geometry for NeRF, IDR, and H-NeRF (from left to right) trained using four cameras.

| Model | Dataset | PSNR ↑ | SSIM ↑ | LPIPS ↓ | Ch $\times 10^{-3}$ ↓ | NC ↑ | IoU ↑ |
|---|---|---|---|---|---|---|---|
| | RenderPeople | 27.33/23.52 | 0.888/0.827 | **0.117**/0.247 | 0.536/0.63 | 0.908/0.892 | 0.864/0.824 |
| NeuralBody [36] | GHS3D | 24.7 | 0.829 | 0.236 | 0.79 | 0.887 | 0.81 |
| | PeopleSnapshot | 24.62 | 0.849 | 0.160 | – | – | – |
| | Human3.6M | 24.86 | 0.82 | 0.189 | – | – | – |
| | RenderPeople | **28.78/24.31** | **0.913/0.856** | 0.125/**0.246** | **0.217/0.274** | **0.950/0.939** | **0.917/0.9** |
| **H-NeRF (ours)** | GHS3D | **24.92** | **0.852** | **0.232** | **0.218** | **0.932** | **0.89** |
| | PeopleSnapshot | **26.33** | **0.868** | **0.159** | – | – | – |
| | Human3.6M | **25.01** | **0.83** | **0.17** | – | – | – |

Table 1: Quantitative evaluation of dynamic sequences for various datasets. When two metrics are reported they correspond to evaluation of training poses and novel poses for test cameras, respectively. PeopleSnapshot and Human3.6M are evaluated on novel poses under training views (where ground-truth images exist). Geometric metrics are only reported when ground-truth is available.

respectively, with a skip connection to the middle layer and with Swish activation [38]. As in the original NeRF [29], we use 256 coarse and fine-level ray samples, for each of which we use 8- and 1-dimensional positional encoding for $F_\omega$ and $\Delta H_\omega$, respectively. We train the network using the Adam optimizer with a learning rate of 0.001 exponentially decayed by a factor 0.1 until the maximum number of iterations is reached ($10k$ iteration of $4k$ ray batch size). We apply Gaussian noise with $\sigma = 0.1$ to the NeRF conditioning code ($\boldsymbol{\theta}, \mathbf{T}, \mathbf{v}$).

**Dataset and Metrics.** We provide qualitative and quantitative evaluation on four different datasets: RenderPeople (8 sequences), GHS3D [51] (14 sequences), PeopleSnapshot [3] (7 sequences) and Human3.6M [19] (5 sequences). The first two are synthetic, rendering animated (RenderPeople) or 4D human scans (GHS3D) from four orthogonal cameras, where ground-truth geometry paired with images is available. We evaluate both the image and the 3D geometry reconstruction quality from two novel cameras, and qualitatively demonstrate generalization to shapes and poses not in the training set. The remaining datasets are real captured videos (PeopleSnapshot: monocular, Human3.6M: four cameras) without paired 3D geometry, where we only evaluate the rendered images. For image metrics, we adopt peak signal-to-noise ratio (PSNR ↑), structural similarity index (SSIM ↑) and learned perceptual image patch similarity (LPIPS ↓) [57]. For evaluation, we render the NeRF field inside the human bounding box $\mathbf{B}$ and compare to the ground-truth segmented image within a region of interest around the person. To evaluate geometric reconstruction quality, we report bi-directional Chamfer (Ch) $L_2$ distance, Normal Consistency (NC) and Volumetric Intersection over Union (IoU), evaluated on the mesh produced by applying Marching Cubes on the SDF, with a resolution of $256^3$.

**Static Human Reconstruction under Sparse Viewpoints.** H-NeRF learns a geometrically regularized volumetric radiance field constrained by an imGHUM-based SDF, which significantly improves robustness under sparse training camera views. In fig. 2, we show reconstruction of a static Render-People and GHS3D scan, respectively, using the original NeRF [29], the state-of-the-art multi-view reconstruction approach IDR [53], and H-NeRF, for increasing number of cameras. Both novel view image quality and geometric accuracy improve for all methods as the number of training views increases. However, in contrast to competitors, H-NeRF produces good quality output even under sparse training views (2-8), demonstrating the generalization capability of a co-training approach.

**Dynamic Human Reconstruction and Rendering.** In the following, we evaluate H-NeRF's capabilities to reconstruct and render dynamic scenes (fig. 5). We compare H-NeRF against NeuralBody [36]

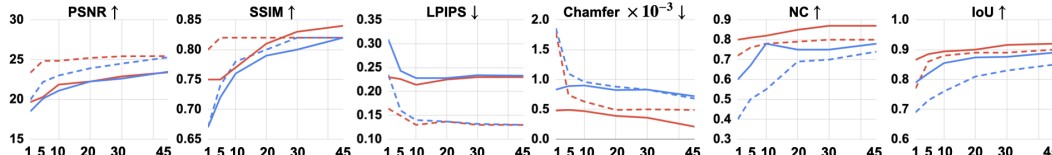

Figure 3: Frame Number Ablation. H-NeRF (red, x-axis: #training video frames) outperforms Neural Body (blue), evaluated on a RenderPeople (dashed) and a GHS3D (solid) sequence.

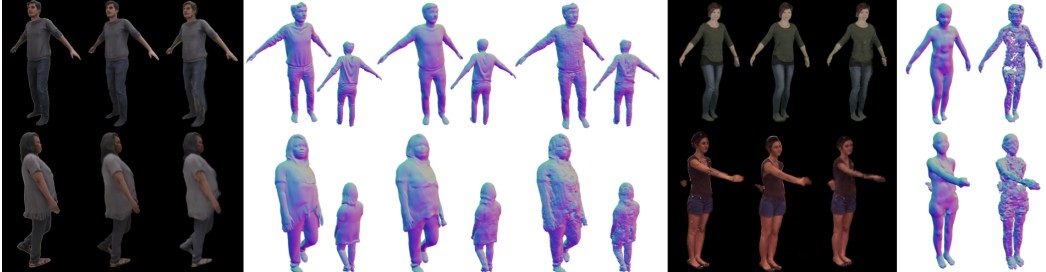

Figure 4: Qualitative comparisons with NeuralBody on four different datasets. Left: GHS3D (top) and RenderPeople (bottom) ground-truth image, our result, NeuralBody, ground-truth geometry (front and back), our geometry, NeuralBody's geometry. Right: PeopleSnapshot (top) and Human3.6M (bottom) ground-truth image, our result, NeuralBody, our geometry, NeuralBody's geometry. Pay attention to the sharp renderings and the complete and detailed geometry produced by our method. Digital zoom-in recommended.

(tab. 1), the current state-of-the-art for novel view synthesis of humans in motion. For fair comparisons, we have trained NeuralBody with the same data as H-NeRF and use GHUM meshes (instead of SMPL). GHUM meshes with the same pose and shape representation as in our model are used for NeuralBody's structured latent codes. We further compare with Nerfies [35]. However, Nerfies has difficulties with the high range of motion in our test sequences and fails for most of them. We consider such results less meaningful and we report them only in the Sup. Mat. for completeness.

H-NeRF overperforms NeuralBody across datasets and metrics, especially for geometric reconstruction. One possible explanation for NeuralBody's lower performance on view synthesis, is that NeuralBody is not conditioned on the root transformation. Thus, global lighting effects are handled less well. Furthermore, pose depended effects that go beyond body articulation are only indirectly modeled through relative distances of input mesh vertices. Finally, NeuralBody conditions on the frame index, which is set to zero for novel poses. On the PeopleSnapshot dataset where no view-dependent and only subtle pose-dependent effects are present, these limitations are not entirely apparent. In contrast, H-NeRF pays special attention to modeling both detailed geometry and appearance, and its explicit conditioning on pose and the root transformation captures pose-dependent geometric and appearance effects. This strategy pays off, especially in more complex scenarios.

**Frame Number Ablation.** We have illustrated H-NeRF's rendering and reconstruction capabilities for static and dynamic sequences. Next, we evaluate the necessary number of different poses or time-frames H-NeRF needs during training for good pose generalisation. We again compare against NeuralBody. To this end, we have trained both methods with an increasing number of images, uniformly distributed over the full sequence. Results are shown in fig. 3. As expected, both methods perform better when trained with more data. However, H-NeRF performs overall better and more importantly, the quality degrades less in the sparse training regime, especially for geometric reconstruction. Our results support H-NeRF's robustness to sparse training data and its capability to reconstruct accurate geometry. In practical terms, H-NeRF can be robustly trained with as little as 10 temporal frames per camera (in a four camera set-up), resulting in a performance capture effort of only a few minutes.

**Qualitative Results for Pose Generalisation and Shape Extrapolation.** Finally, we illustrate H-NeRF's rendering and reconstruction accuracy qualitatively (fig. 4). Consistently with the reported metrics, our synthesized novel poses appear sharper, more detailed, and contain less noise than the current state-of-the-art. The estimated geometry is complete, smooth, and contains much of the detail present in the original scan, e.g. the clothing folds on the back of the person on the left, the first row.

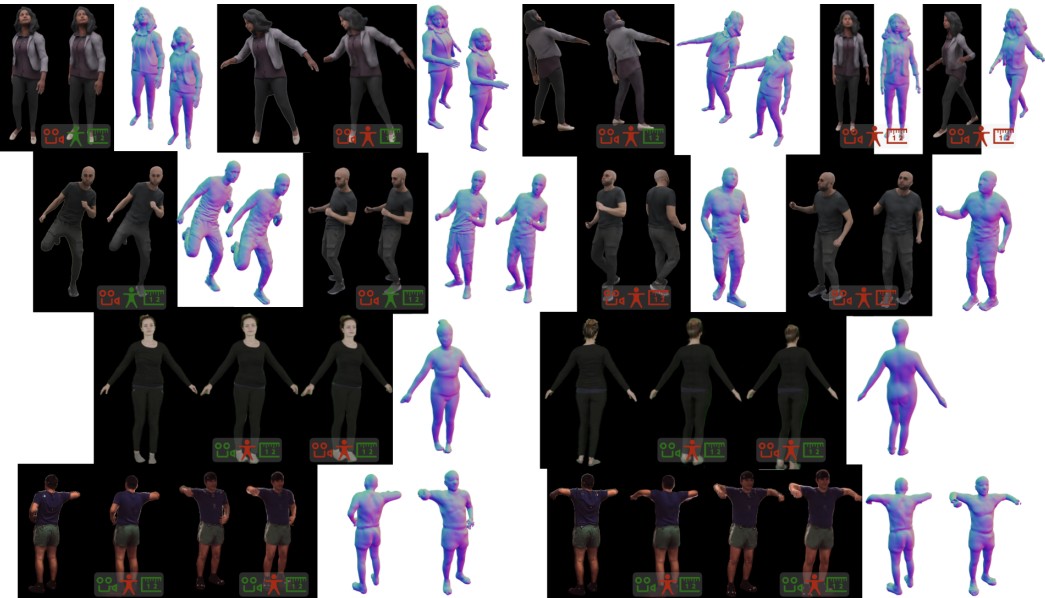

Figure 5: Qualitative results of H-NeRF for novel image synthesis. We show ground truth images and scans (left, if available) side-by-side with our results (right). Red icons correspond to test, green icons to training configurations. Symbols correspond to camera, pose, and shape, respectively. From top to bottom: RenderPeople, GHS3D, PeopleSnapshot and Human3.6M.

In contrast, geometries produced by NeuralBody are noisier and sometimes incomplete. To further demonstrate the versatility of our approach, in fig. 5 we show examples of synthesized images and reconstructed geometry from novel viewpoints, in novel poses, and for modified body shape.

**Limitations and Broader Impact.** In the current setup, we assume full-body views of a single person in every-day clothing. Inconsistency with these assumptions, e.g. by placing another person in the scene and occluding the performer, would break the method although models of partial views, or representation estimates of multiple people could be used for generality. Furthermore, our method exhibits some sensitivity to the estimated body shapes and poses, as well as the quality of image segmentation. Although the process of learning to render could absorb certain inaccuracies, as pose, shape or segmentation are increasingly degraded, the method would fail eventually.

Our method paves the way towards immersive AR and VR applications. In contrast, our approach is not targeted, or particularly useful for applications like visual surveillance or person identification as a particular set-up and a cooperating subject is needed. We do not build an audio-visual model that would be necessary for deep fakes.

## 6 Conclusions

We have presented novel neural radiance field models (H-NeRF) for the photo-realistic rendering and the temporal reconstruction of humans in motion. Our objective is to extend the scope of the previous state of the art, focused on static scenes observed by a large number of cameras, by building dynamic models, which based on a sparse set of views, can generalize well to novel camera views, human body poses, and extrapolate over human body shapes. Our key contribution is in developing a new model with associated multiple losses, in order to specialize and constrain a generic NeRF formulation by using a compatible implicit statistical 3D human pose and shape model, represented using signed distance functions. Our implicit geometric formulation captures not just the statistical regularities of the human body, but also hair and clothing represented as an implicit residual network. Our model is trained end-to-end based on several novel losses, and achieves good results for both 3D reconstruction and photorealistic rendering. Training the body model in an end-to-end rendering framework carries the promise to learn complex implicit skinning functions based on images only, a performance previously possible only in the realm of human capture using complex and expensive 3D body scanners, in the laboratory. We illustrate the favorable capabilities of H-NeRF through extensive experimentation using several datasets and against other state of the art techniques.

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
