# OpenReview forum: "H-NeRF: Neural Radiance Fields for Rendering and Temporal Reconstruction of Humans in Motion"
_NeurIPS.cc/2021/Conference — NeurIPS 2021 Spotlight_

### Official Review · Reviewer_TFXE · 2021-07-15

**Rating:** 8
**Confidence:** 4

**Summary:**

This paper proposes a method for dynamic human reconstruction and rendering using sparse view videos.
The high-level strategy is to learn both a neural radiance field and an SDF field with mutual constraints in a canonical frame from temporal observations. Moreover, for generating live rendering and surface results, pose and position conditioned SDF offsets are also learned for better live frame surface reconstruction results.
To guarantee robust surface reconstruction, an implicit generative human model was used as the inner layer which not only incorporates strong human shape constraint, but also provides semantic correspondences for accurate warping between live frames and the canonical frame. Another benefit of using an underlying implicit human model is that shape and pose editing can be achieved through interpolating in the parametric body space, which was not demonstrated in previous works.

**Ethical Concerns:**



**Limitations And Societal Impact:**



**Main Review:**

Reconstructing/Encoding dynamic humans from light-weight inputs is a very challenging but important topic that has strong potential usage in AR/VR, telepresence, etc.

The paper writing is clear and easy to follow.

The overall idea of the paper is novel, the results in the paper look nice, the quantitative comparisons are impressive.

One of my concerns is that the video results are not well demonstrated, almost all the video results are not very continuous. For example, in video 04:17, the side view rendering results are simply skipped, and this happens to all the results of the PeopoeSnapShot dataset. Moreover, for RenderPeople dataset results in video 01:27, it's easy to get continuous frames for training and rendering, but the videos results are not very continuous, please demonstrate full video results without skipping any frames.

Moreover, I think video comparisons with NeuralBody should be provided to enhance the effectiveness of the proposed method on free-viewpoint rendering and dynamic geometry reconstruction results.

Minor issue:
The caption of Fig.4: botton -> bottom

**Time Spent Reviewing:**

3

---

> ### Author Response · Authors · 2021-08-09
> **Response to Reviewer TFXE**
>
> We are delighted to see that Reviewer TFXE recommends clear acceptance and attests “impressive” comparisons. In the sequel we comment on questions and raised concerns by TFXE.
>
> **Frame sampling**
>
> We did not exclude frames from the RenderPeople motion. The stuttering motion at 01:27 pointed out by Reviewer TFXE was partially because of (1) the abrupt motion change present in the original MoCap sequence (walking dog sequence from Human3.6M); and (2) our frame sampling strategy where we sample MoCap frames every 16th frame to demonstrate a large variety in poses. We will provide smoother video results in the final video with higher temporal sampling rate.
>
> Similar to NeuralBody, we rely on good pose estimation for consistent integration of 2D observations across multiple dynamic frames. Because of degraded monocular pose estimation, we excluded a small number of side view frames from the PeopleSnapshot dataset to maximize the model quality. Our process of fine-tuning imGHUM pose and shape (Eq. 11) helps alleviate the problem (see our ablation study in Tab 4 sup. material) and any other better pose estimation technique (not part of our contribution and not a focus of this paper) would be complementary to our approach. Note that for fair comparisons, we trained NeuralBody with the same images and poses as our method. We will discuss the model dependence on pose estimation accuracy in more detail, and show results that are trained and tested on the full PeopleSnapshot sequences in our final manuscript.
>
> **Further results**
>
> It is easier to identify our improvements over NeuralBody in static frames. However, we agree that adding video comparisons for NeuralBody is a good idea and we will include qualitative video comparisons to the final version.

---

### Official Review · Reviewer_vuz9 · 2021-07-16

**Rating:** 7
**Confidence:** 4

**Summary:**

The authors propose a model that combines a parameteric human shape model with NeRF and a residual signed distance function to enable novel-view synthesis of humans in motion, assuming full observation of the subject (i.e., no prior-based completion).

The core contributions is the combination of NeRF with a parametric human body model, which enables the learning of the NeRF in a canonical coordinate frame, as well as regularization of the learned geometry. Several additional tweaks improve the model further, such as leveraging auxiliarly losses via a instance segmentation mask, or feeding the estimated body pose to the NeRF as additional input, enabling dynamic (due to being pose-specific) fine detail.

**Limitations And Societal Impact:**

I would have appreciated an analysis of runtimes, see "Evaluation" above.

The authors adequately discuss the societal impact of their work.

**Main Review:**

*Related work*
The related work is great.

*Exposition*
The exposition is generally good, however, I found the methods section hard to parse. In particular, I found it very confusing that the authors first discuss the case 4.1 *without* using the coordinate frame established by the imGHUM model. After reading the introduction and the background, I had expected that the authors would leverage imGHUM to establish a canonical coordinate frame that is used to train the NeRF in. Instead, the authors start out explaining a vanilla NeRF without any notion of dynamics. I assume that this was done to set up the scene structuring and residual SDF sections, but personally, I believe the pitch would actually get easier if the authors directly established that the input to the NeRF and the SDF will be the imGHUM's (s, d) tuple. Maybe the authors want to reconsider the narrative of this section.

Instead of "learned", I would recommend the authors call views in the training set "context" or "training" views - "learned" is confusing.

A minor detail: I would recommend the authors remove the audio track of their video, as they did not record any voiceover, but one can still hear static noise and voices in the background.

*Evaluation*
I was glad to see a comparison to NeRF and IDR in the static case, and then a comparison to both NeRFies and neural bodies in the dynamic case.

However, I am not content with the video results. First, in the "Novel View Synthesis" section, I would expect to see the nearest neighbor context camera view, not only the ground-truth for novel camera views. Further, I would like to see what the training set for these sequences looks like - i.e., a mosaic of the context videos before each of the novel view synthesis results.

Further, I would like to see novel camera *trajectories*, not only novel camera *views*.

Only with the previous two results is it possible to judge the multi-view consistency and extrapolation capabilities of the proposed approach appropriately.

Next, I would like to see a discussion of both the rendering and inference times of the proposed approach, which I expect to be very, very slow. What FPS can be achieved at what resolution? How long does it take to train the network?

*Final review*.
I recommend this paper for acceptance. Though I would like to see more comprehensive results in the form of continuous camera trajectories and the context videos, the method is sound and I am convinced that this method performs the way the authors claim it does. The only "shortcoming" of the proposed method is its specificity to the problem at hand, i.e., it is not a fundamental contribution that would enable new approaches across applications in vision, for instance. In particular, this method is quite specific to dynamic novel view synthesis where a parameter-based model is available, which is the case for faces and bodies, but not for general objects. Nevertheless, the contributions are clear, the results are cool and the paper is well written, and I believe this to be a valuable contribution to the community.

**Time Spent Reviewing:**

5

---

> ### Author Response · Authors · 2021-08-09
> **Response to Reviewer vuz9**
>
> We are glad to see that Reviewer vuz9 attests our method to be sound and recommends our paper for acceptance. Below are our comments on the raised questions and concerns.
>
> **Computation/memory consumption/training time**
>
> We trained the models with 10k iterations of 4k ray batch size on 8 Nvidia v100 GPUs, which takes about 6-8 hours for each dynamic sequence. For an image of 512x512 resolution, the inference for H-NeRF on a single Nvidia v100 GPU takes about 9.1 sec whereas the original NeRF takes about 6.5 sec (1.4x faster). As correctly pointed out by the reviewers, the main computation overhead comes from the imGHUM warping, which also limits the maximum query points to be 64**3 due to memory constraints (i.e., 1024 rays consisting of 128 coarse + 128 fine samples). However, we utilize the coarse scene structuring such that we only query imGHUM for points inside the 3D bounding box $\mathbf{B}$ whereas for exterior points (static background or freespace) we use its original position coordinate and a constant distance value (1.0) as input features to the NeRF network. In practice, this significantly reduces the number of imGHUM query points by 90% and largely alleviates the memory constraints. We will add these metrics and additional technical details into the final paper. For future work, one could further improve performance by utilizing the imGHUM SDF for a more compact and efficient ray sampling strategy (for example, as proposed in UNISURF [Oechsle et al. 2021]) or using more recent NeRF variants.
>
> **Camera set-up**
>
> We already show the camera set-up used during training and testing in 1:10 of our sup. video. However, we agree that we could repeat this information before every example for clarity. In our synthetic datasets, the test cameras are “between” two training views. Training views are set up every 90 degrees along the azimuth of a sphere. The test views are offset by 45 degrees such that the azimuth difference is maximized. Global pose and camera changes may introduce similar renderings since both are fundamentally similar. To this end, we fixed the cameras for evaluation clarity. In the final video, we will show sequences where we pause the motion and render static poses under a moving camera.
>
> **Narrative**
>
> We thank the reviewer for the thoughtful comment on our narrative. As suggested, we will aim to better guide the reader by detailing the overall pipeline and clarifying how we compose the various components in overview paragraphs. We will also revise our wording for training/context views.
>
> **Targeted problem**
>
> While focusing on reconstruction of dynamic human scenes, we would like to emphasize “learning high-fidelity dynamic virtual avatars is an important problem” that “has strong potential usage in AR/VR, telepresence, etc” as acknowledged by Reviewer VaSk, MJGq and TFXE.

---

### Official Review · Reviewer_MJGq · 2021-07-17

**Rating:** 7
**Confidence:** 4

**Summary:**

The authors introduce H-NeRF, a geometry-aware neural radiance field for human performance rendering.
The core ideas are to (1) conditioned the neural radiance field (NeRF) on a pre-trained implicit geometric human representation that captures a wide variety of body poses and shapes, and (2) co-learn a signed distance function (SDF) that describes the surface geometry of the NeRF for rendering. Experimental results show that, compared to other baselines, H-NeRF achieves better surface reconstruction and novel view and pose synthesis quality, and are able to extrapolate to new body shape volumes.


**Ethics Review Area:**

["I don’t know"]

**Limitations And Societal Impact:**

The authors have discussed a few limitations in line 326-332. It would be nice if the authors can also clarify the questions/weaknesses raised above.


**Main Review:**

I feel positive about this work. Learning high-fidelity and virtual avatars with animation capability is an important problem. The authors approach this problem with a novel formulation that combines NeRF with human geometric priors, and the empirical evaluations show promising results.

Strength:
Good generalization to unseen poses, comparing to other NeRF-based approaches (e.g., D-NeRF, NeuralBody).
Better surface reconstruction with fewer extraneous points/densities.
Can handle moderate cloth/body shape deformation.
Mostly consistent geometry across different poses.
Can learn from a sparse set of cameras.

Weakness:
Computation/memory consumption/training time is not discussed, please list and add.
Potentially slow inference speed: NeRFs/Implicit functions are often slow, and the presented work has 3 of them in H-NeRF, so I expect the computation time will be higher than other approaches.
Sensitivity to hyperparameters is not discussed: how sensitive H-NeRF is to the weights of the nine different losses? How are the weights chosen? While the weight configuration is in the supplementary (line 89), it would be better if the authors could provide more insights into these losses. Similarly, how are \sigma_h and \gamma determined?
Similar to other human NeRF, H-NeRF needs to fit one model for each human subject.

Below are some further questions and suggestions of how to improve the evaluation:
How does H-NeRF without the residual SDF work (e.g., simply train a NeRF conditioned on imGHUM)?
How does H-NeRF work in extreme poses (e.g., break dancing)?
(line 316, Figure 3) it would be nice if the authors can provide some renderings for comparison. Sometimes small PSNR improvements could make a huge difference perceptually.

Some minor issues:
line 91: citation for NARF [1] is incorrect.
supp. line 96: subjectquetly -> subsequently.

[1] Atsuhiro Noguchi, Xiao Sun, Stephen Lin, Tatsuya Harada. Neural articulated radiance field. arXiv preprint arXiv:2104.03110, 2021.


**Time Spent Reviewing:**

5h

---

> ### Author Response · Authors · 2021-08-09
> **Response to Reviewer MJGq**
>
> We are happy to see that Reviewer MJGq is positive about our work and acknowledges several strengths compared to previous methods. We will cite the paper, and comment on the raised concerns and questions as follows:
>
> **Computation/memory consumption/training time**
>
> We trained the models with 10k iterations of 4k ray batch size on 8 Nvidia v100 GPUs, which takes about 6-8 hours for each dynamic sequence. For an image of 512x512 resolution, the inference for H-NeRF on a single Nvidia v100 GPU takes about 9.1 sec whereas the original NeRF takes about 6.5 sec (1.4x faster). As correctly pointed out by the reviewers, the main computation overhead comes from the imGHUM warping, which also limits the maximum query points to be 64**3 due to memory constraints (i.e., 1024 rays consisting of 128 coarse + 128 fine samples). However, we utilize the coarse scene structuring such that we only query imGHUM for points inside the 3D bounding box $\mathbf{B}$ whereas for exterior points (static background or freespace) we use its original position coordinate and a constant distance value (1.0) as input features to the NeRF network. In practice, this significantly reduces the number of imGHUM query points by 90% and largely alleviates the memory constraints. We will add these metrics and additional technical detail into the final paper. For future work, one could further improve the performance by utilizing the imGHUM SDF for a more compact and efficient ray sampling strategy (for example, as proposed in UNISURF [Oechsle et al. 2021]) or using more recent NeRF variants.
>
> **Hyper-parameters**
>
> The loss weights were first validated empirically using one sequence. We then used the same set of parameters for the rest of our experiments (except differences for $\eta$ for synthetic and real datasets as discussed in the paper). We achieved good results without additional individual parameters fine-tuning. Please acknowledge that a comprehensive hyper-parameter study in 9+ dimensional (+ learning rate, batch size, etc) space is challenging, given that model training is also expensive. However, we did provide an ablation study for the individual losses in Tab. 4 of the supplementary material. $\sigma_h$ is carried over from NeRF. $\gamma$ is determined using grid search as in [Genova et al. CVPR 2019]. However, new findings of other authors and some initial experiments by us suggest that $\gamma$ could be trainable, see VolSDF [Yarif et al. 2021].
>
> **H-NeRF without residual SDF**
>
> Training H-NeRF without the residual SDF could be an interesting ablation study to be added to the final manuscript. However, even with co-training the residual SDF, we already observe the surfaces extracted from the NeRF opacity field are typically less smooth and contain some geometric artifacts (as also pointed out in the UNISURF paper). Removing the residual SDF means training the model without our coupling losses Eq 7, 8, mask loss Eq 9, and geometric regularization Eq 10. As evidenced in our loss ablation study (Tab 4, sup. material), this would further amplify the artifacts in the geometric reconstruction and likely for the novel view synthesis.
>
> **Further results**
>
> For our RenderPeople dataset, we have selected random and quite different motion sequences for training and testing from the CMU and Human3.6M MoCap datasets to demonstrate the strong generalization ability to novel poses. We will add renderings with even more extreme poses, and qualitative results for Figure 3 to the paper/supplementary material.

---

> > ### Comment · Reviewer_MJGq · 2021-08-25
> >
> > Thanks for the review. My questions have been clarified. It would be great to include the runtime info as well as video comparisons with NeuralBody suggested by another reviewer.

---

> > > ### Author Response · Authors · 2021-08-31
> > > **Response to Reviewer MJGq**
> > >
> > > Thank you! We have already incorporated the valuable feedback from all reviewers, and revised the paper and supplementary video accordingly.

---

### Official Review · Reviewer_VaSk · 2021-07-19

**Rating:** 6
**Confidence:** 5

**Summary:**

This paper presents a new NeRF based method for rendering and reconstruction of humans observed from sparse cameras. The main contribution is combining volumetric radiance fields and an implicit SDF for the tasks of novel view synthesis and geometric reconstruction for humans. To achieve this goal, the method utilizes a human prior, imGHUM, which was pretrained on thousands of 3D scans.

**Ethical Concerns:**

No ethical concerns

**Limitations And Societal Impact:**

1. The authors claimed that the key contribution was a new model with associated multiple losses to constrain a NeRF formulation by using the imGHUM statistic model represented using SDF. To me, the losses are not novel, either the regular losses for training NeRF or training neural SDF (i.e. imGHUM in this paper). Moreover, the way to unifying neural radiance fields and SDF is just replacing volume densities with SDF values, more importantly, the issue of having the geometric errors caused by this replacement has not been discussed. Therefore, the technical contributions sound limited.
2. The quality of the novel view synthesis results is limited.
3. The comparisons with the following methods on surface reconstruction are expected:
PIFuHD: Multi-Level Pixel-Aligned Implicit Function for High-Resolution 3D Human Digitization
ARCH: Animatable Reconstruction of Clothed Humans
Video-Based Reconstruction of 3D People Models
Learning to Reconstruct People in Clothing from a Single RGB Camera
4. The comparisons with the following methods on novel view synthesis are expected:
ANR: Articulated Neural Rendering for Virtual Avatars
4. For the novel pose synthesis experiment, more challenging poses are expected to test. For example, poses from the AMASS dataset and the AIST++ dataset.
5. The comparison of IDR in the experiment of static human reconstruction is not fair. For a fair comparison, the prior encoded in imGHUM should also be used in IDR since imGHUM should not be regarded as a contribution of this work.
6. I understand the paper aims at sparse input settings. However, it's good to see the results (qualitative and quantitative results) of IDR and the proposed method when the camera views are sufficient.

**Main Review:**

I like the idea of unifying explicit and implicit representations for rendering as well as reconstruction.
I also appreciate the important problems that the proposed method attempts to address, i.e. photo-realistic free-viewpoint rendering, accurate geometric reconstruction, as well as synthesis for new poses and new shapes.
However, I am not quite satisfied with the proposed solutions and the quality of the results.
For example, the main contribution is to combine NeRF and SDF. The solution is just replacing the density value in volume rendering with a function of the SDF value. However, they did not realize this naive replacement would introduce some geometric errors in the reconstruction results, which was discussed in UNISURF. That is, the formulation of the color accumulation equation in volume rendering would make the point that contributes most to the ray color not lie on the surface. How did the authors solve this issue?
Furthermore, the authors claimed that they can synthesize photo-realistic rendering, however, the novel view synthesis results are blurry and do not meet the standard of photo-realism.
The authors also claimed they can synthesize for new poses, however, the new poses are very simple and thus it is hard to see the generalizability of this method on novel poses. To evaluate the generalizability of this method, the proposed method should be tested on the poses that are very different from training poses, e.g. poses from the AMASS dataset and the AIST++ dataset.

**Time Spent Reviewing:**

3 hours

---

> ### Author Response · Authors · 2021-08-09
> **Response to Reviewer VaSk**
>
> We are pleased to see that Reviewer VaSk – while being critical – generally likes the main idea of our paper and agrees that the problems we address are important. Below are our responses to the points raised by Reviewer VaSk. First we would like to clarify several misunderstandings.
>
> **Clarifications**
>
> Different from UNISURF [Oechsle et al. 2021], we do *not* replace the density of the volume rendering with the signed distance values in the image formation. Instead we *co-learn* and *couple* the SDF and the volume density, respectively. We still rely on the classical NeRF image rendering based on volume density. Eq 7 and 8 are the coupling losses to softly enforce geometric compatibility between the NeRF volume density field and the co-learned SDF. The losses essentially regularize pixel colors to come from the surface. While rendering is performed using volume density, the surface geometry is extracted from the SDF which we further constrain using image masks (Eq 9) and geometric regularization (Eq 10) in order to penalize geometric deviations from the image observations.
>
> We do not unify an explicit and an implicit representation -- we co-learn two related implicit representations, namely NeRF and SDF.
>
> **Novelty of losses**
>
> [1.] We stress that our coupling losses (Eq 7, 8) and the coarse scene structuring losses (Eq 4, 5) are, to the best of our knowledge, novel. Beyond that, we also use established losses to further frame the problem. We demonstrate the usefulness of the individual losses in our ablation study (Tab 4, sup. material).
>
> **Quality of results**
>
> [2.] While naturally leaving room for improvement in future work, we kindly disagree that the “quality of the novel view synthesis results is limited”. As attested by Reviewer MJGq and TFXE, we have achieved visual results of good quality. Our results are clearly better than state-of-the-art as demonstrated e.g. in Table 1, Fig 3, and Fig 5.
>
> **Comparisons**
>
> [3., 4.] Thank you for pointing out related work on human reconstruction. We could add comparisons to PIFuHD, ARCH, and [Alldieck et al. 2019] if requested by the reviewers. However, we believe these methods are targeted at partially related but quite different problems: PIFuHD and ARCH are single image reconstruction methods trained using a large corpus of dressed scans. Both methods do only a single inference step – in contrast, we  train a subject specific network on multiple frames. [Alldieck et al. 2019] is limited to A-poses and also depends on a large corpus of dressed scans for pre-training. Their paper can be conducted for visual comparison as they also use the PeopleSnapshot dataset. All such methods have in common the objective to reconstruct a static (sometimes LBS-skinned) shape, whereas we reconstruct a dynamic scene with view- and pose-dependent effects. In fact, NeuralBody [35] compares to some of these methods and we demonstrated that our method is qualitatively and quantitatively better than [35]. ANR was not published at the time of our submission (appeared on CVPR 2021) and the code is not publicly available even now. We will cite all methods in our final paper.
>
> [6.] The comparison with IDR is fair in the sense that the intent of the comparison is to show that using a geometric prior is an advantage given sparse views. Combining a geometric prior with IDR is not pursued by any prior work and would defeat the point of this comparison. We do not claim imGHUM to be a contribution of this paper. Our contribution is to integrate imGHUM and NeRF into a consistent framework for learning and reconstruction of subject specific dynamic scenes given sparse views.
>
> **Further results**
>
> [5.] We will add results for more extreme poses to the final version of the paper. However, we note that (1) we already use a large variety of poses from the CMU dataset (subset of AMASS) and Human 3.6M, and (2) that train and test sequences are randomly selected and are already quite different (e.g. walking dog vs. gestures).
>
> [7.] We already show quantitative results of IDR/NeRF for sufficient views in Fig. 2. We will add qualitative results to the supplementary material. As already evidenced in the metrics, H-NeRF and NeRF converge to very similar results while showing some differences with IDR due to the different image formation process.

---

> > ### Comment · Reviewer_VaSk · 2021-08-30
> >
> > Thank you for the response and clarification!
> > I have a question regarding the benefit of co-training the volume density and SDF. One drawback of IDR is that it requires mask supervision to make the training converge to a valid surface. I expected that the proposed method in this work could solve this problem, however, it seems the training of the proposed method still needs foreground masks as supervision? Then what is the benefit of introducing the co-training strategy? Even without this co-training strategy, IDR seems to work well if the foreground masks are given.
> > Another related question is whether it is possible to integrate imGHUM and IDR directly. If it is possible, I would like to see this result.

---

> > > ### Author Response · Authors · 2021-08-31
> > > **Response to Reviewer VaSk**
> > >
> > > Thank you for your questions.
> > >
> > > **imGHUM + IDR**
> > >
> > > Please notice that for our models, we pursue high-quality for *both* the geometric reconstruction and the view synthesis capabilities. As shown in our *static* scene comparison (fig.2), with sufficient views (16+), IDR converges to worse image quality than NeRF (see plots for PSNR, SSIM, LPIPS), even though it shows comparable surface reconstruction capability (plots for Chamfer, NC, IoU). We therefore prefer volume rendering (NeRF) over IDR for better view synthesis and co-train a residual SDF for high-quality geometric reconstruction using only a sparse set of views (1-4). Our observations have also been confirmed in UNISURF (Introduction, paragraph 3 & 4 and fig. 2 & 3).
> > >
> > > As far as we can tell, there is no obvious, or straightforward way to integrate imGHUM and IDR. For example, IDR requires SDF tracing to identify the surface. Thus, imGHUM alone would not be sufficient to represent the geometry as it is not modeling clothing or hair, even in the static case. IDR has only been developed for *static* scene reconstruction. Applying IDR to *dynamic* scenes or combining it with a geometric prior could possibly be interesting, but is not pursued by any prior work. It is hence an open research problem to integrate IDR and imGHUM into a consistent pipeline, and to formulate adequate training losses for rendering and reconstruction of dynamic human scenes. imGHUM + IDR is a completely new system and could be a paper on its own. Concluding, we do not see any obvious imGHUM+IDR baseline that can be easily performed as a comparison for this paper.
> > >
> > > **Co-training residual SDF**
> > >
> > > As mentioned before, we are pursuing *both* realistic view synthesis *and* high quality geometric reconstruction. We argue that novel-view rendering and reconstruction are two sides of the same coin, and reliable viewpoint generalization, especially given relatively few input views, would require good quality for both. The residual SDF captures all details of the subject that are not currently modeled by imGHUM, such as hair or clothing. Without our residual SDF, one could choose to extract surfaces from the NeRF opacity field. However, even *with* co-training the residual SDF, we already observe that surfaces extracted from the NeRF opacity field are typically less smooth and contain some geometric artifacts (as also pointed out in the UNISURF paper). Removing the residual SDF means training the model without our coupling losses Eq 7, 8, mask loss Eq 9, and geometric regularization Eq 10. As evidenced in our loss ablation study (Tab 4, sup. material), this would further amplify the artifacts in the geometric reconstruction and likely for the novel view synthesis.
> > >
> > > **Mask supervision**
> > >
> > > As shown in our paper (Eq. 5, 7, 9), we do use foreground masks for image supervision, scene structuring and geometric regularization. Similar to NeuralBody, we specifically focus on the geometric reconstruction and rendering of the dynamic *human* and we do not focus on learning the background. We emphasize that our problem setting is very different compared to UNISURF which only models *static* scenes with abundant camera views and tries to reconstruct the full scene including background. One could still train our H-NeRF without mask supervision. The model will then synthesize images of people, including background. However, we have no geometric prior for the background and it is impossible to generalize well to novel views given only very sparse training views (single view in the extreme, or 4 in our many other experiments). In addition, we consider mask supervision as a very useful image signal in practice–as demonstrated in our loss ablation study (Tab 4, sup. material), it enhances both geometric reconstruction and novel view synthesis.

---

> > > > ### Comment · Reviewer_VaSk · 2021-09-01
> > > >
> > > > Thank you for the reply! The authors addressed most of my concerns. I would like to raise my rating.

---

### Decision · Program_Chairs · 2021-09-27

**Decision:**

Accept (Spotlight)

**Comment:**

The submission has received 4 positive final ratings: 6, 7, 7, 8.
The reviewers overall were excited about the method and the idea of combining implicit representations with explicit priors, and also acknowledged strong empirical results and solid presentation. The remaining questions and concerns were mostly addressed in the rebuttal: the reviewers were mostly satisfied, but left some recommendations for further improvements (the authors are recommended to follow them while preparing the camera ready version).
The final recommendation is to accept as a spotlight.